# Structural basis for the transport and regulation mechanism of the multidrug resistance-associated protein 2

Eriko Koide [1], Harlan L. Pietz [1,2], Jean Beltran[3] & Jue Chen [1,4] ✉

Multidrug resistance-associated protein 2 (MRP2) is an ATP-powered exporter important for maintaining liver homeostasis and a potential contributor to chemotherapeutic resistance. Using cryogenic electron microscopy (cryo-EM), we determine the structures of human MRP2 in three conformational states: an autoinhibited state, a substrate-bound pre-translocation state, and an ATP-bound post-translocation state. In the autoinhibited state, the cytosolic regulatory (R) domain plugs into the transmembrane substrate-binding site and extends into the cytosol to form a composite ATP-binding site at the surface of nucleotide-binding domain 2. Substrate displaces the R domain, permitting conformational changes necessary for transport. These observations suggest that the R domain functions as a selectivity gauge, where only at sufficiently high concentrations can the substrate effectively initiate transport. Comparative structural analyzes of MRP2 bound to various substrates, as determined in this study and others, reveal how MRP2 recognizes a diverse array of compounds, supporting its role in multidrug resistance.

Multidrug resistance-associated protein 2 (MRP2), also known as ABCC2, is a member of the ATP-Binding Cassette (ABC) transporter family. It is expressed in the liver, intestines, kidney, and placenta[1,2]. Patients deficient in MRP2 function develop Dubin-Johnson Syndrome, a disease caused by an excess of the bile pigment bilirubin in the liver cells[3–5]. Mutations that decrease the amount of functional MRP2 have also been associated with greater incidence of toxic liver injury and hepatotoxicity during administration of cytotoxic drugs[6–9]. MRP2 knockout mice exhibit hyperbilirubinemia, reduced bile flow, reduced biliary glutathione excretion, and an increase in liver size[10]. These data indicate that MRP2 plays a role in maintaining liver homeostasis by excreting potentially toxic molecules.

In addition to endogenous substrates such as bile components and conjugated metabolites, MRP2 also recognizes and transports xenobiotic molecules including anthracyclines and vinca alkaloids[1]. MRP2 upregulation is associated with poor prognosis in several cancers, likely arising from the ability of MRP2 to transport many chemotherapeutic molecules[11,12]. MRP2 inhibition is associated with improved efficacy of cisplatin sensitivity in hepatocellular carcinoma, and MRP2 knockout mice exhibit decreased excretion of the chemotherapeutics methotrexate and doxorubicin[10,13]. Hence, MRP2 poses a persistent challenge in the pharmacology of multidrug resistance-associated proteins: while inhibition holds therapeutic promise to overcome drug resistance in chemotherapy, it also carries the risk of inducing toxicity.

MRP2 is a single polypeptide comprising of an N-terminal transmembrane domain (TMD0) and the canonical transporter core of two transmembrane domains (TMD1 and TMD2) and two nucleotide-binding domains (NBD1 and NBD2) (Fig. 1A). The function of TMD0 in MRP2 remains unclear and is only conserved in a small number of ABC transporters. Deleting TMD0 in a homologous protein, MRP1 (also known as ABCC1), had no functional effects on ATP hydrolysis and substrate transport[14,15]. The transporter core, on the other hand, is highly conserved among ABC exporters. In these exporters, the two TMDs form a substrate translocation pathway, while the NBDs bind and hydrolyze ATP[16]. Additionally, MRP2 contains a 100-residue long

[1]Laboratory of Membrane Biology and Biophysics, The Rockefeller University, New York, NY, USA. [2]Weill Cornell/Rockefeller/Sloan Kettering Tri-Institutional MD-PhD Program, New York, NY, USA. [3]Department of Biology, Davidson College, Davidson, NC, USA. [4]Howard Hughes Medical Institute, The Rockefeller University, 1230 York Ave, New York, NY, USA. ✉e-mail: juechen@rockefeller.edu

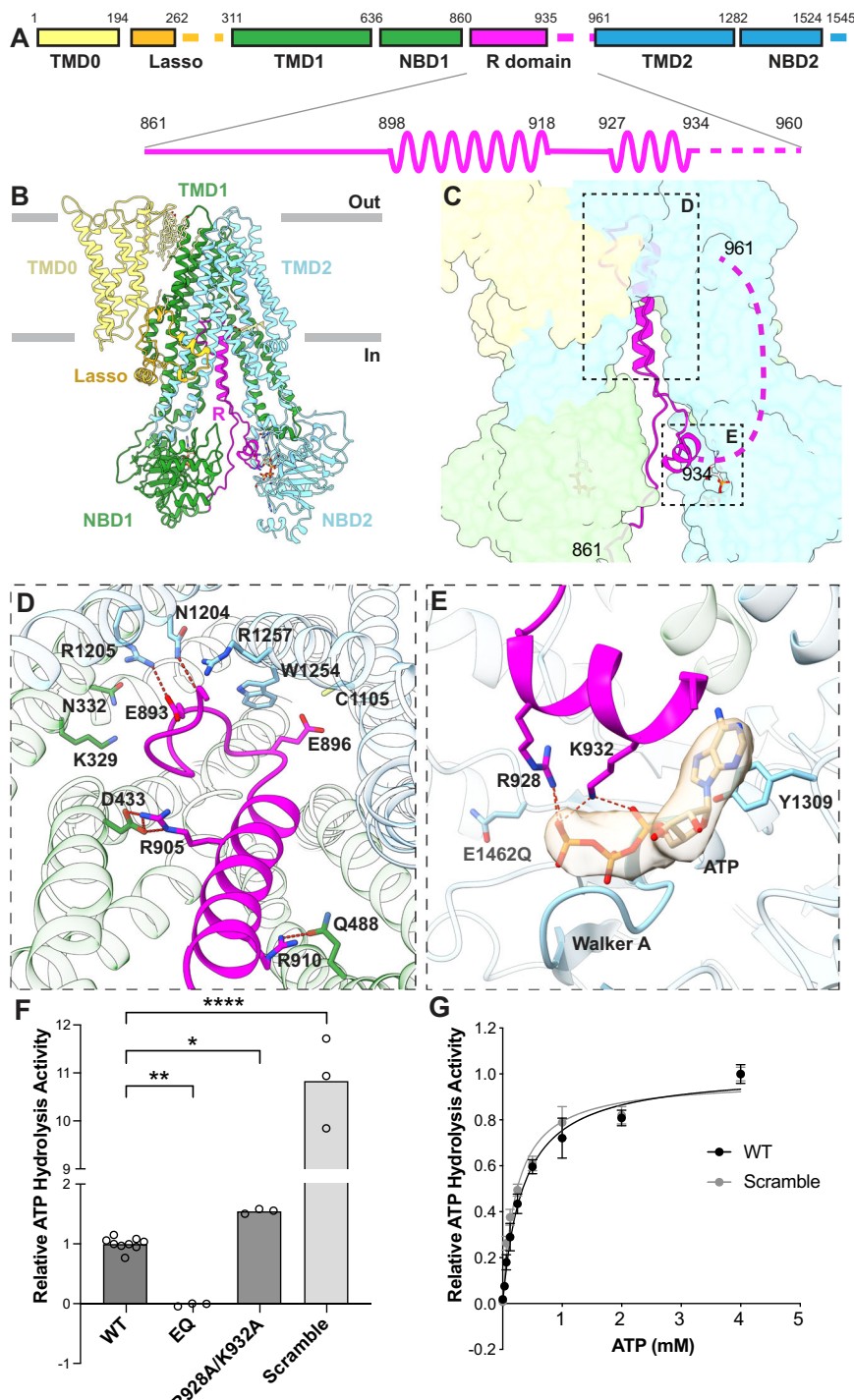

**Fig. 1 | The resting state of MRP2 reveals an auto-inhibitory R domain. A** Domain organization of MRP2 with the secondary structure of the R domain depicted. Dashed lines indicate unresolved regions. **B** Ribbon representation of the structure of MRP2(E1462Q) in the inward-facing, ATP-bound conformation. Domains are colored as in (**A**). **C** Zoomed-in view with TMDs and NBDs represented as surfaces, colored as in (**A**) and (**B**), and the R domain in ribbon. Boxes indicate the positions of the two interfaces illustrated in panels (**D** and **E**). (**D**) R domain inserts into the substrate-binding site as viewed from the cytoplasm. Hydrogen bonds are indicated by red dashed lines. Colored as in (**B**). **E** An ATP molecule binds between the R domain (magenta) and NBD2 (blue). Cryo-EM density of ATP (contour level 0.08) is depicted as a tan transparent surface. Dashed lines indicate hydrogen bonds. **F** Relative ATPase activity of MRP2 variants. WT: wild type MRP2, EQ: the E1462Q variant, R928A/K932A: double mutation; and Scramble: a variant in which the R

domain sequence is randomly scrambled. ATP hydrolysis rates were measured in the presence of 4 mM ATP/magnesium at a protein concentration of 75 nM, except for the E1462Q variant, which was 500 nM. Bar height represents the mean value of three separate measurements (unfilled dots) normalized to the average of WT activity. Statistical significance was calculated using the matched mixed-effects model and Dunnett's multiple comparisons test on measured ATPase activity values (not normalized), comparing the mean of each variant to WT. The $p$ value of MRP2(E1462Q) is 0.0011, for R928A/K932A is 0.0105, and for Scramble is < 0.0001 (*$P < 0.05$, **$P < 0.005$, ***$P < 0.0001$). **G** Relative ATPase activity of MRP2 WT and Scramble variants. Each datapoint represents the mean of 6 replicates, with ATPase activity normalized to the average activity at 4 mM ATP. Error bars represent standard deviation. The curve was fitted with the Michaelis-Menten least squares fit.

linker between NBD1 and TMD2, resembling the regulatory (R) domain of the cystic fibrosis transmembrane regulator (CFTR).

A fundamental puzzle regarding multidrug resistance is how a single transporter can recognize many unrelated drugs. The mechanisms of other multidrug transporters, including P-glycoprotein (P-gp, ABCB1), ABCG2, and MRP1 have been well-studied. For example, P-gp and ABCG2 both contain a central hydrophobic cavity that many small-molecules, including substrates and inhibitors, can bind largely through van der Waals interactions[17–22]. The substrate-binding site in MRP1 is very different from those in P-gp and ABCG2, and is accessible only from the cytoplasm. It can be divided into two parts: a positively charged region containing many polar functional groups (which was termed the P-pocket) and a largely hydrophobic area capable of interacting with different hydrophobic moieties (the H-pocket)[14]. In addition, many residues in the binding site exhibit side-chain plasticity that would allow for adaptability to a variety of substrates. The mechanism by which MRP2 can recognize many substrates with different chemical structures remains to be elucidated.

In this study, we use cryogenic electron microscopy (cryo-EM) to investigate the structural dynamics of MRP2 throughout its transport cycle. Through structural analysis of MRP2 in auto-inhibited, pre-translocation, and post-translocation states, we observed large scale conformational changes enabling ATP-dependent substrate translocation. Furthermore, by comparing the structures of MRP2 in complex with different substrates, we begin to understand how a single transporter recognizes a diverse array of substrates.

## Results

### The resting state of MRP2 is an auto-inhibited state

We first determined the structure of the wild type (WT) human MRP2 in the absence of substrate and ATP (apo) and observed an inward-facing, NBD-separated conformation (Figs. S1, S3A). We also analyzed the structure of a hydrolysis-deficient variant (E1462Q) in the presence of 5 mM ATP, which exhibited two conformations with approximately equal populations (Fig. S2): an inward-facing, NBD-separated conformation and an NBD-dimerized conformation which we will discuss later. Despite the different sample preparation conditions, the inward-facing conformations of the WT and the E1462Q variant were very similar: when superimposed, the overall root mean square deviation (RMSD) is 3.1 Å for 1422 Cα positions (Fig. S3B, C). Because the structure of the E1462Q variant reveals more complete density of the R domain, we will discuss the inward-facing conformation based on this structure (Fig. 1).

In this conformation, the structures of TMD0 and the transporter core are both well defined. While the five TM helices of TMD0 constitute a compact domain, the twelve TM helices of TMD1 and TMD2 form two domain-swapped, pseudo symmetric bundles, each with an NBD attached in the cytosol (Figs. 1B, S3C). Interactions between TMD0 and the transporter core occur mostly in the extracellular loops and with the lasso motif (Figs. 1B, S3D). In the transmembrane region, TMD0 and the transporter core have few direct contacts, with densities likely representing cholesterol and lipid molecules filling the gap between them.

A unique feature of MRP2, compared to other MRP molecules, is its structured R domain—the 100-residue linker between NBD1 and TMD2 (Figs. 1B, C, S2C). The N-proximal half of the R domain (residues 861–893) forms a long loop along the inner surface of NBD1, extending nearly 90 Å into the TM cavity (Fig. S3E). Residues 894–897 form a sharp turn at the top of the TM cavity, and residues 898–918 form an α-helix between the two TM bundles (Fig. 1C, D). The remainder of the R domain extends into the cytosol and forms a short helical segment at the inner surface of NBD2 (Fig. 1E). Density for the C-terminal 25 residues of the R domain was absent, indicating the flexibility of the linker connecting to TMD2.

The position of the R domain suggests an inhibitory role in substrate binding and ATP hydrolysis. Near the apex of the TM cavity, the loop-turn-helix motif of the R domain engages in hydrogen bonding and extensive van der Waals interactions with TMD1 and TMD2 (Fig. 1D). Many residues at this interface, including R1205, N1204, R1257, W1254, and N332, are involved in substrate binding (discussed later, in Fig. 2). Additionally, the helical region of the R domain forms contacts with both halves of the transporter, through a salt bridge formed between R905 and D433 in the first half and a hydrogen bond between R910 and Q488 of the second half (Fig. 1D). This configuration is reminiscent of the inhibited structure of the peptide transporter TAP, in which a viral inhibitor ICP47 forms a helical hairpin inserting into the intracellular openings of the transporter to preclude substrate binding and conformational change necessary for transport function[23,24].

In addition, the C-proximal region of the R domain forms a composite ATP-binding site in conjunction with the Walker A/B motifs of NBD2 (Fig. 1E). An ATP molecule is positioned between these two distant sequences and is stabilized by the canonical ATP-binding residues in NBD2, as well as by hydrogen bonds with R928 and K932 in the R domain (Fig. 1E). This configuration, which to date has not been observed in any ABC transporter, suggests a mechanism for regulating transport activity. Like other members of the ABCC subfamily, MRP2 possesses only one catalytically competent ATPase site. This site is formed between the Walker A/B motif of NBD2 and the LSGGQ motif of NBD1 when the NBDs are dimerized. By sequestering the ATP-binding site in NBD2, the R domain of MRP2 competes directly with the signature sequence of NBD1, adding a new layer of control.

To assess the functional role of the R domain, we generated two variants: an R928A/K932A double mutant, which disrupts ATP coordination at the interface with NBD2 but retains substrate binding-site interactions, and a scramble variant, in which the entire R domain (residues 861–960) is replaced with a scrambled amino acid sequence. Compared to the wild-type protein, the double mutant caused a small but reproducible increase in basal ATPase activity. The scramble variant, which likely disrupts most or all interactions with the transporter core, led to a more pronounced effect. Basal ATPase activity increased by more than 10-fold, while the Michaelis constant (Km) for ATP decreased modestly from $0.35 \pm 0.03$ mM (WT) to $0.23 \pm 0.02$ mM (mean ± standard error) (Fig. 1F, G). These results suggest that the change in ATPase activity for the scramble variant is largely due to the increased rate of NBD dimerization rather than a difference in affinity for ATP. The milder increase in ATPase activity observed with the R928A/K932A double mutant suggests that basal ATPase activity in MRP2 is likely originated by the occasional release of the R domain to permit NBD dimerization, rather than by ATP hydrolysis at the ATP-binding site on NBD2 in the NBD-separated conformation.

### Substrate-binding alleviates auto-inhibition

To understand how substrate is recognized by MRP2 and the effect of substrate binding on the autoinhibitory R domain, we determined the cryo-EM structure of WT MRP2 in the presence of leukotriene C4 (LTC4), a pro-inflammatory signaling molecule that is transported by multiple MRP transporters (Figs. 2, S4). Cryo-EM reconstruction obtained in the presence of 40 μM LTC4 revealed the same structure as the auto-inhibited state; only in the presence of 285 μM LTC4 were we able to observe clear density for LTC4 in the substrate-binding site (Figs. 2E, S4E). The cryo-EM map, refined to an overall resolution of 2.75 Å, shows no density for the R domain, indicating that the R domain is flexible in this conformation. These results suggest that only at sufficiently high concentrations can the substrate displace the R domain to be transported.

LTC4 binds at the interface of the two TM bundles, triggering a global conformational change that brings the two halves of the transporter core closer together (Figs. 2B, S5A). The inward-swing of the

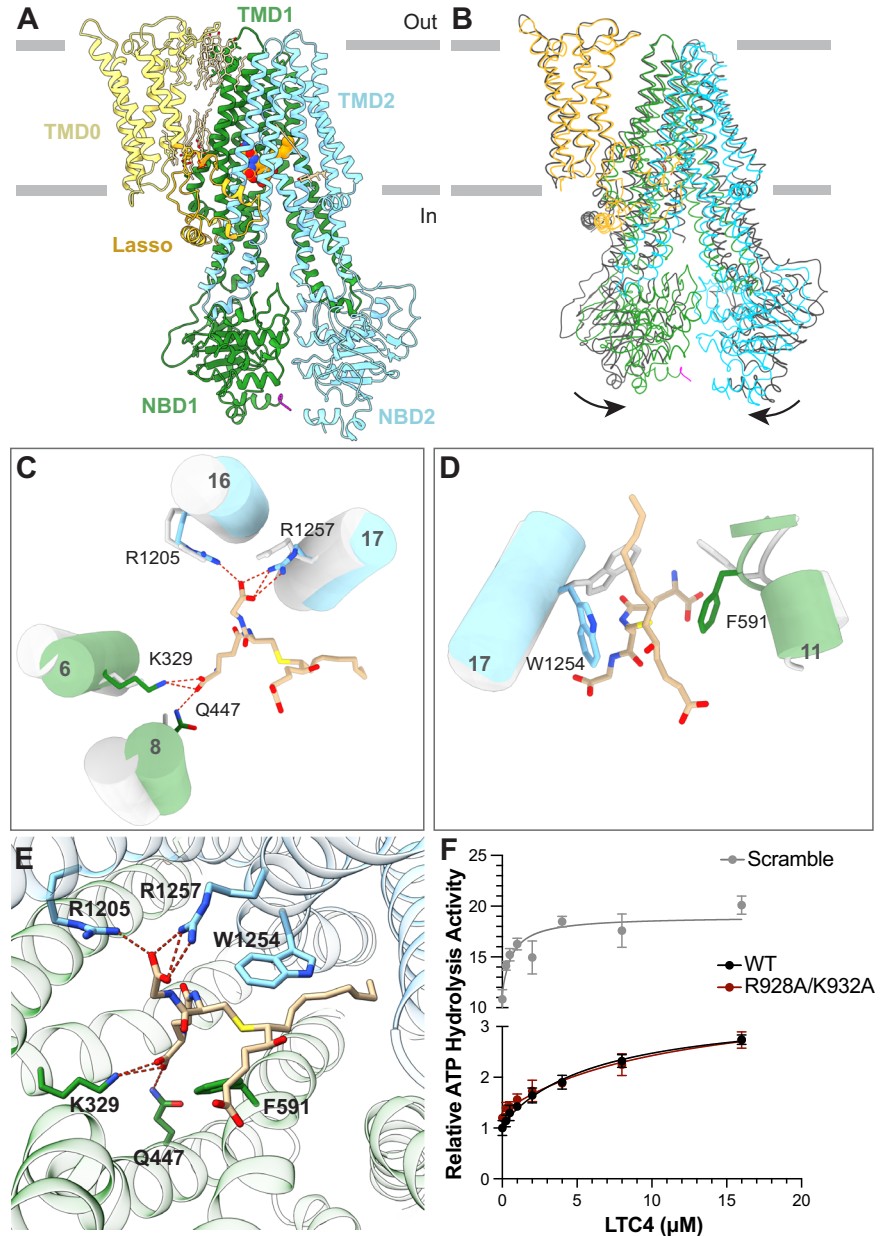

**Fig. 2 | LTC4 binding releases R domain inhibition. A** The overall structure of the LTC4-bound MRP2, colored as in Fig. 1B. **B** Superposition of the LTC4-bound structure, colored as in (**A**), with that of the substrate-free MRP2 (gray). Arrows indicate the inward movement upon LTC4 binding. **C**, **D** Local conformational changes at the LTC4-binding site, TMD1 in green, TMD2 in blue, LTC4 in tan, overlayed with inward-facing, substrate-free MRP2 (gray). Dashed red lines represent hydrogen bonding. **E** Molecular details of LTC4 binding MRP2, viewed from the cytosol. Colored as in (**C**), with TMD1 in green, TMD2 in blue, LTC4 in tan, and hydrogen bonds as red dashed lines. **F** ATP hydrolysis activity of MRP2 variants, normalized to the basal activity of wild type MRP2 at saturating ATP concentration. Each datapoint represents the mean value of 3 replicates. Error bars represent standard deviation. The curve was fitted with the quadratic binding formula for substrate binding (see "methods"). The $K_m$ values for LTC4: WT: $6.3 \pm 1.5$ μM, R928A/K932A: $11 \pm 4$ μM, and Scramble: $0.64 \pm 0.28$ μM (mean ± standard error).

NBDs originates at the substrate-binding site, where TM helix 6, 8, 16, and 17 move towards each other and the side chains of F591 and W1254 reposition to contact LTC₄ (Fig. 2C, D). Previous studies of MRP1 have shown that LTC4 induces a similar conformational change and accelerates the transition of MRP1 from an NBD-separated, inward-facing conformation to an NBD-dimerized, outward-facing conformation[25,26]. Given that LTC4 also stimulates the ATPase activity of MRP2 (Fig. 2F), it is plausible that MRP2 operates through a comparable mechanism.

Although LTC₄ interacts with both MRP1 and MRP2, its affinity for MRP2 is approximately ten times lower than that for MRP1[27]. This difference can be attributed to structural differences at their respective binding sites, as well as the presence of the R domain in MRP2

(Fig. S5B). In MRP2, LTC₄ is largely stabilized by hydrogen bonds between the GSH moiety and residues K329, Q447, R1205, and R1257 (Fig. 2E). Despite the high resolution of the cryo-EM map and clear density for the GSH moiety, density for the lipid tail is mostly absent except for a short portion stabilized by van der Waals interactions with W1254 (Figs. 2E, S4E). In comparison, in MRP1, the GSH moiety of LTC4 forms an additional hydrogen bond with Y440. The significance of this interaction is underscored by the Y440F variant of MRP1, which leads to reduced ATPase activity upon LTC4- and GSH-stimulation[28]. In MRP2, the corresponding residue to Y440 is, in fact, a phenylalanine (F437), mirroring the Y440F variant of MRP1. In addition, the hydrophobic tail of LTC4 in MRP1 is stabilized by

π-orbital stacking interactions with the side chains of M1092, Y1242, and W553, which are not conserved in MRP2 (Fig. S5B). This difference further contributes to the relative lower affinity of MRP2.

LTC$_4$ stimulated the ATP hydrolysis rate of both WT and the two MRP2 variants, but with different potencies (Fig. 2F). The apparent $K_m$ of the WT protein is approximately 6.3 uM. While perturbation of the ATP-binding residues (the R928A/K932A variant) had only a minor effect on the $K_m$ value, disrupting the entire R domain (the scramble variant) increased the apparent affinity for LTC$_4$ by 10-fold. These data suggest that the R domain functions as an affinity/concentration gauge, selecting substrates with either high affinity or those that accumulate to high intracellular concentrations.

### NBD dimerization enables substrate release to extracellular space

The structure of the NBD-dimerized conformation, determined from the dataset collected for the E1462Q variant in the presence of ATP, provides a structural understanding of how substrate is transported and released to the extracellular space. The map is refined to an overall resolution of 3.4 Å (Fig. S2A, E–G) and shows two copies of ATP and magnesium ions at the NBD dimer interface (Fig. S6A, B). Similar to the inward-facing structures, lipid molecules fill the gap to bridge TMD0 and the transporter core (Fig. S6C). Density for the transmembrane helices near the extracellular leaflet is less well defined, suggesting that these helices are relatively flexible (Fig. S2F, compared to Figs. S2C, S1C).

In this conformation, much of the R domain remains unresolved, except for the C-proximal residues 945–960. These residues dock along the intracellular region of the TM helices, stabilized by a network of van der Waals interactions and hydrogen bonds formed between E955 and R1083, as well as between the mainchain atoms of K953 and G960 and the side chains of R1079 and R1150, respectively (Fig. 3A, B). A similar configuration was previously observed in MRP1[25] despite the limited sequence homology between the MRP2 R domain and the MRP1 L1 linker (Fig. S6D). This structural similarity suggests a common role for the linker region in stabilizing the NBD-dimerized conformation.

The TMDs of MRP2 form an outward-facing conformation in which the substrate-binding site is open to the extracellular space and closed off to the cytosol, creating a clear translocation path through which the substrate can be released to the extracellular space (Fig. 3C–F). Several key changes at the substrate-binding site further facilitate substrate release. Upon NBD-dimerization, TM helices 6, 8, and 16 move away from each other, pulling apart residues that interacts with LTC4. Furthermore, R1257 forms a cation-π interaction with W1254, stabilizing W1254 in a conformation that would sterically clash with the substrate (Fig. S6E). These conformational changes have a net effect of reducing the affinity for LTC4, thereby facilitating its release to the extracellular space.

## Discussion

In this manuscript, we present three structures of human MRP2 to elucidate the structural changes associated with its transport cycle and identify a unique autoinhibited conformation. During the preparation of this manuscript, two similar studies were published, with one describing the structure of human MRP2 and the other describing those of rat Mrp2 (rMrp2)[29,30]. Comparing the results obtained in these three independent studies offers a deeper understanding of the MRP2 transport mechanism.

All three studies report a similarly inward-facing, apo conformation with the R domain inserted into the transmembrane substrate-binding site. In addition, we observed a previously unresolved region of the R domain, residues 927–934, reaches onto the NBD2 surface to form a composite ATP-binding site (Fig. 1). Given that MRP2 hydrolyzes ATP with a Km of 0.35 mM and that intracellular ATP concentration ranges between 1 and 10 mM, the ATP-bound conformation observed in this study is likely to represent a physiological state of MRP2. In this conformation, the R domain inserts into the intracellular space between the two halves of the transporter, occupying the substrate-binding site in the TMDs and sequestering the ATPase site on NBD2. Perturbation of the R domain insertion, either by point mutation or sequence randomization, increased the activity of the transporter ([29] and this study), underscoring the functional role of the R domain as an auto-inhibitor. Only at sufficiently high concentrations can substrate effectively compete for the binding site, promoting the disengagement of the R domain from the inhibitory position. Furthermore, Beis and colleagues have reported higher activities of the fully phosphorylated rMRP2 compared to dephosphorylated or partially phosphorylated protein, suggesting that phosphorylation may regulate the transport activity in a mechanism akin to CFTR[30]. Whether and how phosphorylation of the R domain regulates MRP2, and which kinases are involved, will be the subjects of future studies.

Comparison of the NBD-dimerized, ATP-bound MRP2 structure determined in this study with that of ATP/ADP-bound MRP2 described by Mao and colleagues[29] reveals that the two structures are nearly identical, with a RMSD of 1.5 Å between 1385 Cα pairs (Fig. S6F). These two structures represent two conformational states: the pre-hydrolysis state with ATP molecules bound at both ATPase sites, and the post-hydrolysis state in which ATP in the catalytically competent site has already been hydrolyzed. The nearly identical structures of these two conformations indicates that, similar to MRP1[25,26], substrate is released before ATP hydrolysis, and dissociation of the NBD dimer likely constitutes the rate-limiting step in the transport cycle.

Collectively, the three studies support a mechanism of how ATP hydrolysis is coupled to substrate export (Fig. 4). In the absence of substrate, MRP2 rests in an inward-facing conformation with the R domain inserted between the separated NBDs. At physiological ATP concentrations, ATP is likely bound at the interface of NBD2 and the R domain, further stabilizing the R domain in the auto-inhibitory position (Fig. 4A). When the intracellular concentration of the substrate becomes sufficiently high, it can displace the R domain from the transmembrane binding site (Fig. 4B) to permit NBD dimerization (Fig. 4C). The conformational changes in NBDs are transmitted to the TMDs, opening the translocation pathway to the extracellular space and lowering its affinity so that the substrate will be released. ATP hydrolysis followed by ADP release then resets the transporter to its inward-facing configuration, ready again to bind substrate from the cytoplasm.

Comparing the three ligand-bound structures of MRP2 ([29,30] and this study) also provides an opportunity to understand how a single transporter recognizes a spectrum of compounds (Fig. 5). The molecular masses of these three compounds differ by a factor of three, ranging from 285 g/mol (probenecid) to 840 g/mol (BDT). Their chemical structures are also very different: LTC4 is an arachidonic acid derivative conjugated to the tripeptide GSH, BDT is bilirubin conjugated with two taurine moieties, and probenecid is a small sulfonamide. A common feature of these compounds is their bipartite nature, with one portion negatively charged and the other hydrophobic. All three compounds bind to the same site in MRP2, which contains a positively charged P-pocket and a mostly hydrophobic H-pocket. The negative moieties of each compound form hydrogen bonds with a set of highly conserved residues in the P-pocket, including K329, Q447, N1204, and R1205 in TM Bundle 1 and R1257 in TM Bundle 2 (Fig. 5). These hydrogen bonds are critical for ligand recognition, as alanine substitution of a single positively charged residue nearly abolished ATPase stimulation by BDT[29]. In contrast, interactions with the H-pocket vary among the different compounds. The lipid tail of LTC4 is largely unstructured, while in the cases of BDT and probenecid, a cholesterol molecule bridges their interactions with residues in the

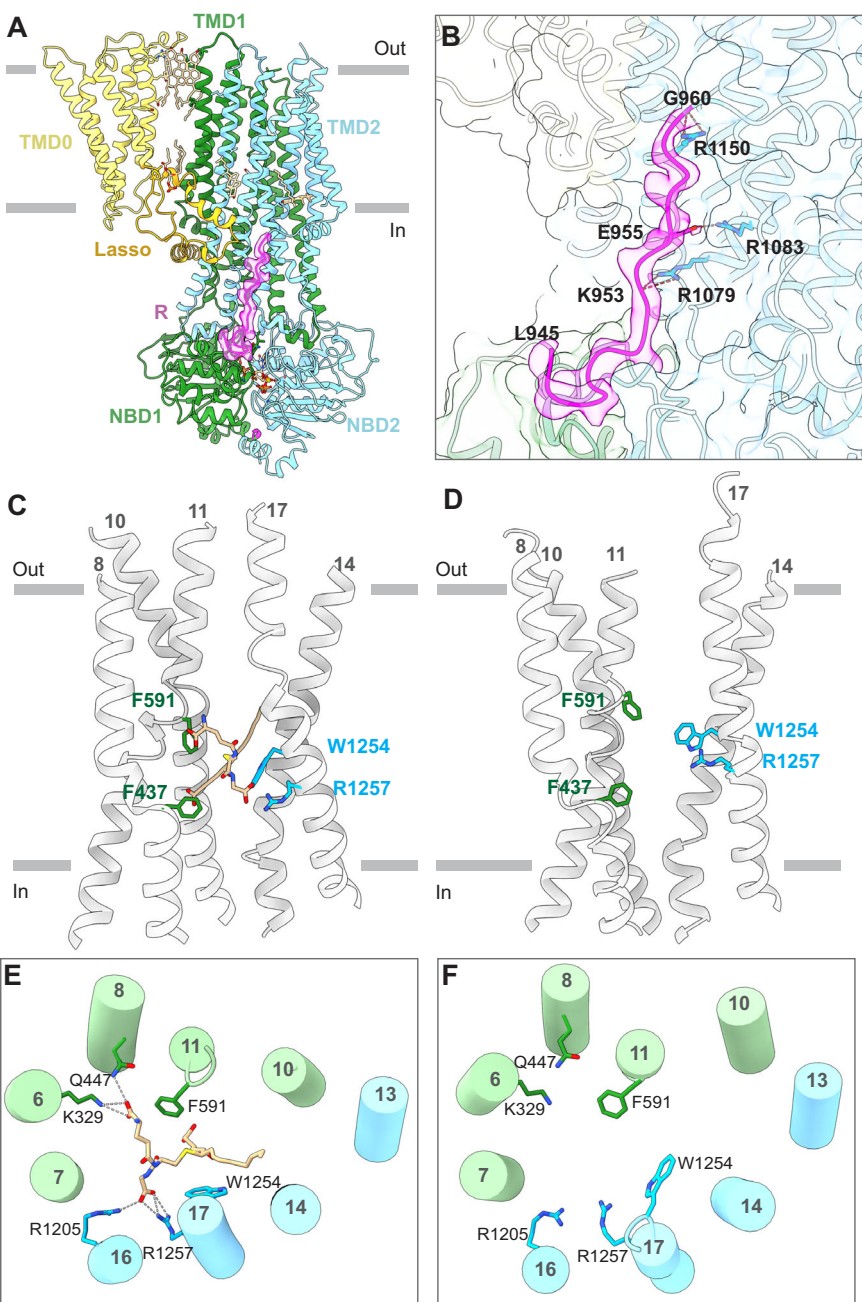

**Fig. 3 | The NBD-dimerized, outward-facing conformation. A** Ribbon representation of the overall structure, colored as in Fig. 1B. Also shown is the cryo-EM density corresponding to the R domain at a contour level of 0.095 (magenta surface). **B** Zoomed-in view of the R domain, colored as in (A). Hydrogen bonds are represented as dashed lines. **C**, **D** conformational changes at the substrate-binding site in the inward-facing, LTC4-bound state (**C**) versus the outward-facing conformation (**D**). Residues in green correspond to TMD1, residues in blue correspond to TMD2. **E** The structure of the substrate-binding site in LTC4-bound conformation, viewed from the extracellular space. TMD1 depicted in green, TMD2 depicted in blue. Hydrogen bonds are represented as dashed lines. **F** The structure of the substrate-binding site in outward-facing conformation, viewed from the extracellular space.

H-pocket. Unlike LTC4 or BDT, Probenecid, a drug used to treat gout and gouty arthritis, is inadvertently exported by MRP2[31]. Despite being much smaller than typical MRP2 substrates, two copies of Probenecid occupy the binding site to facilitate its efflux by MRP2[30]. These data indicate that substrate recognition is primarily driven by hydrogen bonds formed with charged or polar residues in the P-pocket. The H-pocket merely provides a "sticky" surface to accommodate hydrophobic moieties. This configuration enables MRP2 to accommodate different substrates, if they possess a negatively charged moiety capable of binding to the P-pocket and a hydrophobic region compatible with the H-pocket. These data also underscore the promiscuous nature

of these multidrug transporters and the challenge of avoiding drug resistance conferred by them. Identifying the molecular determinants of ligand recognition and unique characteristics of each transporter will facilitate future drug development efforts aimed at overcoming these challenges.

## Methods

### Cell culture

*Spodoptera frugiperda* (Sf9) cells (Gibco) were cultured in Sf-900 II SFM medium (Gibco) containing 5% fetal bovine serum (FBS) (Gibco) and 1% antibiotic-antimycotic (Gibco) at 27 °C. HEK293S GnTI⁻ cells

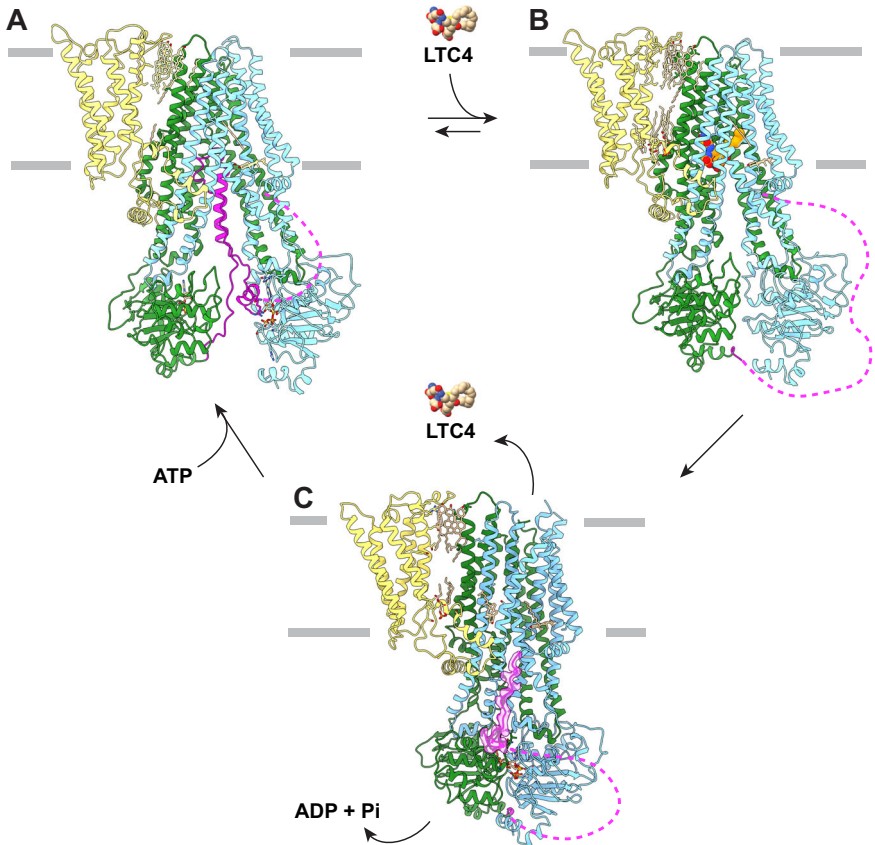

**Fig. 4 | The MRP2 transport cycle. A** In the absence of substrate, MRP2 rests in an auto-inhibited state. Colored as in Fig. 1B. Dotted magenta lines represent unresolved regions of R domain. In this state, ATP-binding does not induce NBD dimerization, but rather stabilizes R domain in the auto-inhibiting position. **B** Substrate displaces R domain from the auto-inhibited conformation. **C** Substrate is released in the NBD-dimerized conformation.

(ATCC) were cultured in Freestyle 293 medium (Gibco) containing 2% FBS and 1% anti-anti at 37 °C, 8% $CO_2$, and 80% humidity. Cell lines were tested for mycoplasma contamination using the Universal Mycoplasma Detection Kit (ATCC) every month.

### Protein expression and purification

Mammalian codon-optimized gene encoding human MRP2 (BioBasic Inc.) was subcloned into a plasmid in frame with a C-terminal PreScission Protease cleavage site followed by an eGFP tag. The vector was then transformed into DH10Bac cells to generate recombinant bacmid. Purified bacmid was transfected into Sf9 cells and the resulting baculovirus was amplified to P4. P4 baculovirus was added at 10% v/v to HEK293S GnTI⁻ cells at a density of 2.5–3.5 million cells/mL. Sodium butyrate was added 12 h later to a final concentration of 10 mM, and the temperature was dropped from 37 °C to 30 °C. Cells were harvested 48 h later, flash frozen in liquid nitrogen, and stored at −70 °C.

For protein purification, cells were thawed and solubilized in buffer containing 300 mM NaCl, 50 mM HEPES pH 8.0 with KOH, 2 mM $MgCl_2$, 2 mM DTT, 20% glycerol, 2% LMNG, 0.2% CHS, 1 μg/mL aprotinin, 0.6 mM benzamidine, 1 mM PMSF, 1 μg/mL leupeptin, 1 μg/mL pepstatin A, 100 μg/mL soy trypsin inhibitor, and DNase I. Insoluble material was removed by centrifugation at 75,000 × g for 40 min. Supernatant was mixed with GFP nanobody-conjugated Sepharose 4 Fast Flow resin (GE Healthcare). Resin was washed with buffer containing 0.06% digitonin, 150 mM NaCl, 50 mM HEPES pH 8.0 with KOH, 2 mM $MgCl_2$, 2 mM DTT, and 5% glycerol. PreScission protease was added to the column and incubated overnight at 4 °C to release MRP2 from the resin. Eluate was passed through GST Sepharose resin (Cytiva)

to remove PreScission protease. The protein was concentrated using a 100 kDa filter to approximately 1 mL and further purified by size-exclusion chromatography at 4 °C using a Superose 6 10/300 GL column (GE Healthcare) equilibrated with buffer containing 0.06% digitonin, 150 mM KCl, 50 mM Tris pH 8.0, 2 mM $MgCl_2$, and 2 mM DTT.

### Mutagenesis

MRP2(E1462Q) and MRP2(R928A/K932A) were generated using PCR amplification with mutagenic primers. These primers were complementary to the template plasmid except at the bases targeted for mutation (Integrated DNA Technologies, IDT). Following PCR amplification of the MRP2 wild type plasmid with these primers, the product was digested by Dpn1 for 4 h (New England Biolabs, NEB). The PCR product was purified by gel purification (Zymo Research). The purified plasmid was added to 50 uL DH5α cells (NEB) and transformed following the manufacturer protocol. Following recovery, 200 uL transformed cells were spread on LB/ampicillin plates and incubated at 37 °C overnight. Clonal plasmid DNA was then expanded, purified, and sequenced for verification (Genewiz).

To generate the Scramble construct, the R-domain amino acid sequence was randomly scrambled and ordered as gene fragments from IDT. The parent MRP2 plasmid was amplified using primers complementary to the regions of MRP2 directly adjacent the R-domain and to the N- and C- termini of the scrambled R-domain. The PCR product of this amplification was purified by gel extraction and assembled with the scrambled R-domain using Gibson assembly (NEB) at a 1:5 g/g and 1:10 g/g ratio. 5 uL of the assembly product was added to 50 uL NEB5a cells, transformed, expanded, and purified as above.

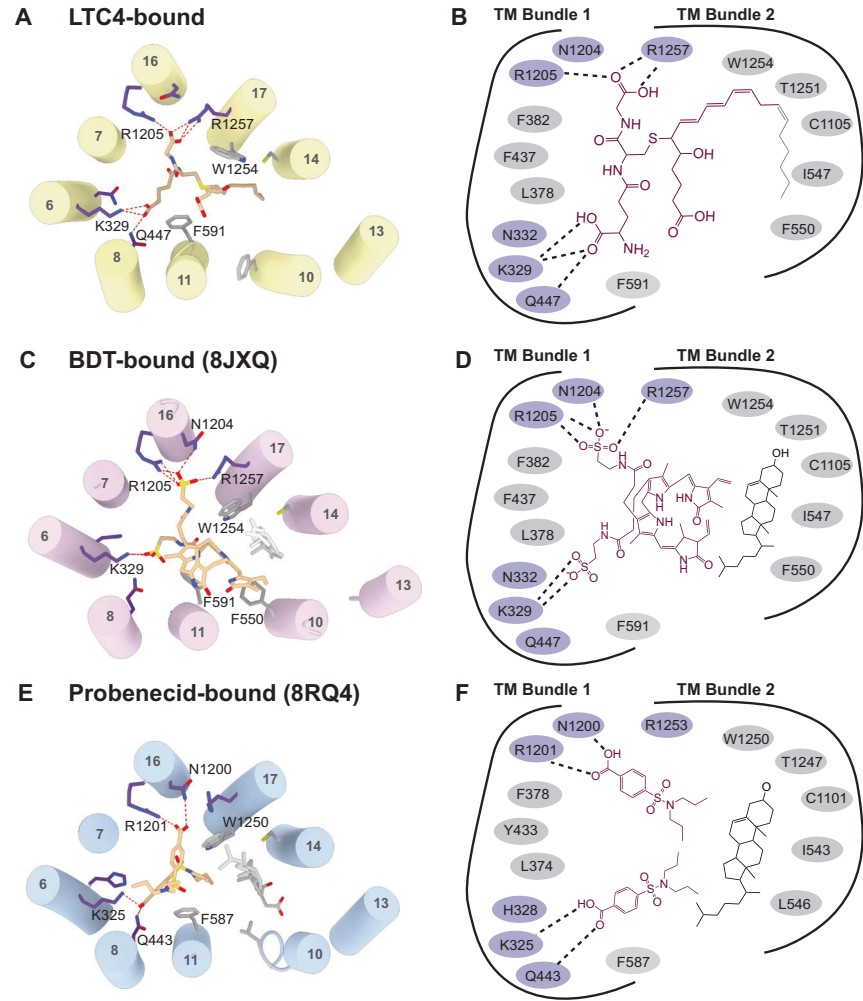

**Fig. 5 | Structural basis of multi-substrate recognition. A** The structure of the substrate binding site observed in the LTC4-bound conformation (yellow). Residues in the P-pocket that interact with LTC4 are indicated. Hydrogen bonds are depicted as dotted red lines. **B** Schematic drawing of the LTC4 binding site. LTC4 is shown in burgundy, with unresolved hydrophobic tail in gray. Hydrogen bonds are depicted as dashed lines. **C** The structure of substrate-binding site observed in the BDT/cholesterol (CLR)-bound conformation (pink)[29]. BDT is depicted in tan, CLR in gray. **D** Schematic drawing of the interactions of BDT (burgundy) and CLR (black) with MRP2 as in (**B**). **E** The structure of binding site observed in the Probenecid/cholesterol-bound rMRP2 (blue)[30]. Probenecid is depicted in tan, cholesterol in gray. **F** Schematic drawing of the rMRP2 bound with probenecid (burgundy) and CHS (black). Portion of CHS which extends away from the substrate binding site residues is not pictured. In panels (**B**, **D**, and **F**) the conserved residues that form hydrogen bonds with substrates are highlighted in purple.

## ATP Hydrolysis assay

ATP hydrolysis activity was measured using the NADH-coupled assay described previously[32,33]. Briefly, MRP2 was added at a concentration of 75 nM to a reaction mixture containing 50 mM HEPES, 150 mM KCl, 2 mM MgCl$_2$, 2 mM DTT, 0.06% w/v digitonin, 60 µg/mL pyruvate kinase, 32 µg/mL lactate dehydrogenase, 9 mM phosphoenolpyruvate, and 150 µM NADH in gel filtration buffer. For LTC4 stimulation assays, LTC4 was added to this reaction buffer at no higher than 1.6% DMSO (DMSO volume was consistent across all assayed wells). 30 µL aliquots were distributed into wells of a black/clear bottom Corning 384-well polystyrene microplate. The reaction was initiated by the addition of 4 mM ATP-Mg$^{2+}$ and NADH depletion was measured by monitoring $\lambda_{ex}/\lambda_{em}$ 340/445 nm using a Tecan Infinite M1000 Pro (Tecan). The rate of NADH depletion was measured from a minimum of three separate measurements per condition and converted to ATP turnover with an NADH standard curve using Prism 10. Data was fit to the quadratic velocity equation, a modification of the Michaelis-Menten equation which accounts for tight-binding substrates.

$$Range\ Normalization\ Factor = V_{max} - V_0 \qquad (1)$$

Where $V_0$ = ATPase activity in absence of LTC4, and $V_{max}$ = ATPase activity in presence of saturating LTC4.

$$v = V_o + (V_{max} - V_o)\,\frac{([E_T] + [S_T] + K_m) - \sqrt{([E_T] + [S_T] + K_m)^2 - 4[E_T][S_T]}}{2[E_T]}$$

$$(2)$$

## Cryo-EM sample preparation

MRP2 Apo (PDB:9C2I): Immediately following gel filtration, MRP2 was concentrated to 4.5 mg/mL (27 µM). 3 mM fluorinated Fos-choline 8 (FFOS8) was added to sample just before application onto glow-discharged Quantifoil R0.6/1 300 mesh Au grids. 3 uL of sample was applied to grids, blotted for 3 s with a blot force of 15 at 100% humidity and 22 °C, and frozen in ethane using a Vitrobot Mark IV (Field Electron and Ion Company). The LTC4-bound structure (PDB: 9C12) was obtained in the presence of 285 µM LTC4 and 2.5% DMSO. The MRP2 (E1462Q) structures (9BR2, 9BUK) were obtained in the presence of 5 mM ATP-Mg$^{2+}$ and 40 µM LTC4.

## Data collection

Cryo-EM images for WT MRP2 apo and MRP2(E1462Q) structures were collected as described previously on a Titan Krios (FEI) using a K3 Summit direct electron detector (Gatan, Inc)[25,32]. Cryo-EM images for the LTC4-bound MRP2 were collected on a Titan Krios (FEI) using a Falcon 4i Direct Electron Detector (ThermoFisher). All micrographs were collected in SerialEM using superresolution mode[34]. Data collection parameters have been summarized in Table S1.

## Image processing

Strategies for data processing have been outlined in Supplementary Figs. 1, 2, and 4. Super-resolution images were corrected for gain reference and binned by 2. Beam-induced motion was corrected using MotionCor2[35]. Contrast transfer function (CTF) estimation was conducted using CTFFIND4[36]. Particles were picked using RELION-3 Laplacian-of-Gaussian autopicking function, binned by 4, extracted in RELION and imported into cryoSPARC[37]. Particles went through several rounds of 2D classification, and resulting particles underwent ab initio reconstruction with 3–4 classes. The best class demonstrated density protruding from the detergent micelle, while other classes resembled empty micelles. These classes were used to conduct heterogeneous refinement on the original particle stack. Nonuniform refinement of the best class following heterogeneous refinement yielded a low-resolution reconstruction of MRP2. The particles from the best class were imported into RELION using the csparc2star.py script, where they were re-extracted without binning and sorted by iterations of 3D classification using the density from nonuniform refinement as the reference map[38]. The best classes were refined by non-uniform refinement and local refinement in cryoSPARC. FSC curves were generated in cryoSPARC. All resolutions are reported at the 0.143 FSC cutoff.

## Model building and refinement

The initial models of each conformation of MRP2 were obtained by docking the previously reported models of the transporter core of bMRP1 (PDB:5UJ9, 6BHU) and the AlphaFold2 model of TMD0 into the sharpened maps and mutating each residue to the corresponding MRP2 residue based on sequence alignment[14,25,39,40]. Models were then manually adjusted into the density using Coot. Models were then refined using ISOLDE and PHENIX, followed by manual adjustment of side chain geometry to best fit the cryo-EM density[41,42]. Regions with poor side chain density were built as poly alanine models. The quality of the models was evaluated by MolProbity[43].

## Reporting summary

Further information on research design is available in the Nature Portfolio Reporting Summary linked to this article.

## Data availability

The cryo-EM maps of Apo MRP2, inward-facing and outward-facing MRP2(E1462Q), and LTC4-bound MRP2 have been deposited in the Electron Microscopy Data Bank under the accession codes EMD-45159, EMD-44833, EMD-44911, and EMD-45099, respectively. The corresponding atomic models have been deposited in the Protein Data Bank under accession codes 9C2I, 9BR2, 9BUK, and 9C12. All other data are available in the main text, supplemental information, or source data for Figs. 1F, 1G, and 2F. The data that support this study are available from the corresponding authors upon request. The following atomic models have also been referenced in this work: 8JXU (Cryo-EM structure of human ABC transporter ABCC2 under active turnover condition). 8JXQ (Cryo-EM structure of bilirubin ditaurate (BDT) bound human ABC transporter ABCC2). 8RQ4 (Cryo-EM structure of the rat Multidrug resistance-associated protein 2 (rMrp2) in complex with probenecid). Source data are provided with this paper.

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

## Acknowledgements

We thank Mark Ebrahim, Johanna Sotiris, and Honkit Ng at the Rockefeller University Evelyn Gruss Lipper Cryo-EM Resource Center and Nick Spellmon at the HHMI Janelia Cryo-EM Facility for their support with collecting electron microscopy data. We thank members of the Chen and Mackinnon laboratories for helpful discussions. This work was supported by HHMI to J.C., the National Institute of General Medical Sciences (T32GM007739 to the Weill Cornell/Rockefeller/Sloan Kettering Tri-Institutional MD-PhD program) and a Predoctoral Fellowship from the National Cancer Institute (F30CA257282) to H.L.P.

## Author contributions

E.K. and J.C. designed research; E.K. performed all the experiments with help from H.L.P., and J.B.; E.K. and J.C. analyzed data; E.K. and J.C. wrote the paper.

## Competing interests

The authors declare no competing interests.
