## [Transparent Peer Review file · Nature Communications]

Structural basis for the transport and regulation mechanism of the Multidrug Resistance-associated Protein 2

Corresponding Author: Dr Jue Chen

Version 0:

Reviewer comments:

Reviewer #1

(Remarks to the Author)

- This manuscript is overall well-written and easy to follow. However, there are major edits required for the work as it stands. The manuscript reads as if it was started before the first human ABCG2 structures were reported, and then later the authors pivoted to discussing their structures along with the previously reported structures. Although there are definitely novel data presented in this manuscript, the authors are not convincing in championing their work as a contribution to the field. For example, the authors report a previously unresolved portion of the R-domain which is absent from previously reported structure which reaches onto the NBD2 surface to form a composite ATP-binding site. This is a really interesting finding, especially given that NBD2 is the consensus site, but it's barely highlighted or discussed. Similarly, the analysis of the binding pocket based on their LTC4-bound structure and previously reported BDT- and probenecid-bound structure is cursory. This manuscript would be greatly strengthened by the elaboration of the discussion and analysis of these two topics.
- Although it is not explicitly required, for consistency naming using the HUGO gene nomenclature is recommended for this manuscript. Currently, the authors use a mix of HUGO and common names for the transporters (MRP2, MRP1, P-gp, ABCG2). The authors do state P-gp is also known as ABCB1, however MRP2 and MRP1 are never defined as ABCG2 and ABCG1, respectively. It is recommended to identify all transporters as their HUGO Gene Nomenclature Committee (HGNC) names, or at the very least to include both.
- There is a bit of a disconnect between the abstract and the manuscript submitted. Line 28 (abstract): "Comparative structural analyses of MRP2 in complex with different substrates reveal how the transporter recognizes a diverse array of compounds, highlighting the transporter's role in multidrug resistance." The wording of this phrase in the abstract suggests that this manuscript will present structures of ABCG2 with different substrates. However, only one substrate-bound structure is presented, and the comparison is performed using previously derived structures from other groups. This needs modification to more accurately reflect what was evaluated in the current manuscript. In a similar vein, none of the substrates bound to ABCG2 in these structures are chemotherapeutics, so it seems more appropriate to say that this manuscript discusses the polyspecificity or promiscuity of ABCG1, which supports its role in multidrug resistance.
- Lines 171-175: "Disruption of the interactions with the substrate-binding site led to a more pronounced change. Replacement of the entire R domain (residues 861-960) with a scrambled amino acid sequence ("scramble") increased the basal ATPase rate by more than 10-fold (Fig. 2G). These results suggest that by disrupting the interactions between the R domain and the transporter core, the R domain is more readily released from its auto-inhibitory role, leading to an elevation in the ATPase activity even in the absence of substrate." Given that the entire R-domain sequence has been scrambled, you have in essence mutated the R-domain to something that is not an R-domain. I agree with your conclusion that the ATPase activity is increased. However, because the R-domain is absent, it seems inconceivable that a completely scrambled version of this domain is still able to properly insert and release from within the TMDs. Please verify this structurally. Also, because the ATPase changes so strongly, please determine if this is due to a change in the NBD dimerization and/or change in affinity for ATP.
- Lines 199-200: "In particular, K953, Q955, and G960 of the R domain form hydrogen bonds with R1079 and R1083 of TM helix 14 and with residue R1150 of TM helix 15." K953 is not shown in Figure 3 and the 955 residue is labeled as E955, not Q955. I believe this is a typo in the text.

- Lines 223-226: "All three studies report a similarly inward-facing, apo conformation with the R domain inserted into the transmembrane substrate-binding site. In addition, we observed the C-proximate region of the R domain that, in the presence of ATP, reaches onto the NBD2 surface to form a composite ATP-binding site (Fig. 1)." This section of the R-domain is unresolved in previously reported ABCC2 structures, correct? It seems like this discrepancy should be discussed more prominently.
- Line 237: "Chen and colleagues" is used for reference 24, which corresponds to the corresponding author, not the first author, Mao. Please modify.
- Line 264 through end of discussion: "The negative moieties of each compound form hydrogen bonds with the same set of residues in the P-pocket (Fig. 5)." The analysis provided in this paragraph is quite broad and vague. These statements would be perfectly appropriate if the binding pocket was discussed more in depth previously, either when discussing the LTC4-bound structure or in a dedicated section preceding the discussion. However, the discussion of the promiscuity of the binding pocket is largely found wanting.
- Methods section: There are multiple instances of a space included between a given number and "°C." Please revise so that, for example, "30 °C" becomes "30°C."
- Line 319-320: "New England Biotechnologies, NEB." I believe NEB stands for New England Biolabs.
- Line 340: " $\lambda_{ex}/\lambda_{em}$ " requires subscripts
- Figure 2D, 5B,D, and F, and S5: Each of these figures includes at least one panel with schematic drawings where amino acid residues in colored ovals are shown interacting with chemical structures of ligands. In figure S5, the coloring is well explained (P-pocket in blue, H-pocket in gray). However, this panel is the only one with adequate details to interpret the figure. Figure 2D shows ovals in blue and green, which I have interpreted as referring to which TMD the residues come from, as is colored in Figure 2A-C. However, the caption must explain what the color scheme means for figure 2D. I am completely lost on Figure 5 B, D, and F, which contain purple and gray ovals. The coloring does not correlate with the numbering as observed in Figure 2D, where TMD1 residues (#329-591) are in green and TMD2 (#1101-1257) are in blue. The coloring also does not correlate with the P-pocket and H-pocket residues from figure S5. Please either update the caption to explain why the residues are colored differently or update the coloring and caption to mirror figure 2D or S5.

Reviewer #2

(Remarks to the Author)

Comments on the manuscript entitled "Structural basis for the transport and regulation mechanism of the Multidrug resistance associated protein 2" by Koide et al.

In this work, the authors reported four cryo-EM structures of human MRP2 in three conformational states: an autoinhibited state for the wild-type in the absence of ligand and ATP (3.6 Å) and for the E/Q mutant in the absence of ligand but in the presence of ATP (3.4 Å), a pre-translocation state for the wild-type with substrate-bound (LTC4) in the absence of ATP (2.9 Å), and an ATP-bound post-translocation state for the E/Q mutant (3.4 Å). The authors reported the observation of large-scale conformational changes that enable ATP-dependent substrate translocation, showing that MRP2 functions through the classic alternating access model, driven by ATP binding and hydrolysis.

The EM density for the regulatory R domain was found to be traceable for a large portion of the domain, leading to a revelation of its two interaction sites with other parts of MRP2. In addition to the structural work, two mutants were made in the R domain and analyzed for their importance to the observed interactions. The R domain was proposed to serve as a selectivity gauge, wherein only sufficiently high concentrations of substrates or substrate with sufficiently high affinity can effectively compete with and disengage the R domain to initiate transport.

The authors performed cryoEM experiments and structure determination carefully. The cryoEM data processing flow charts illustrated in supplementary figures did not show any sign of discrepancy. For example, the FSC curves are acceptable for C1 reconstruction, and the 2D class averages and final reconstructions with local resolution maps are all within expectation.

Although this reviewer recognizes past important contributions of the group to the structural studies of ABC transporters in general and ABCC family proteins in particular, there are several concerns with the current submission.

Major concerns:

1. The results reported here are consistent with the two recent publications in January and February of this year in Nature Communications by Mazza et al. (Reference 24 for the rat MRP2 structures in nucleotide-free autoinhibited state and probenecid bound state) and by Mao et al. (References 25, human wild type MRP2 in apo state, BDT bound state, and ATP/ADP state). In addition to their observation of the insertion of the R domain into the substrate-binding site in the TMD, these two papers also demonstrated regulatory effect of R domain on MRP2's ATPase and transport activities using mutagenesis, phosphomimetics, and proteins that are fully dephosphorylated, partially and fully phosphorylated. Thus, the new information presented in this submission appears to limit to an additional interaction between the C-proximate region of the R domain and the ATP bound at the NBD2 in the IF conformation. To enhance its novelty, the authors should provide

additional data in the manuscript.

2. The title of the manuscript is vague enough to be an umbrella under which this work might be presented if it wasn't for statements like "different substrates reveal how the transporter recognized a diverse array of compounds, highlighting the transporter's role in multidrug resistance" (Abstract lines 29-30). Although the discussion on the structural basis of multi-substrate recognition is interesting, it is based largely on results from the two recent publications.

3. Like the two recent publications, the focus of this manuscript is about the R domain, especially the part of R domain from residues 927-934 that is found in this work interacting with ATP bound at the NBD2. The authors imply that this interaction, in the absence of substrate, prevents MRP2 from hydrolyzing ATP, providing another level of regulation in addition to the universally observed interaction at the substrate-binding site. Therefore, the R domain plays a role as an off switch in resting state in order to conserve ATP in cells. However, this interaction of R domain (927-934) with ATP bound in NBD2 can only be observed in the E1462Q mutant instead of WT, calling into question whether this interaction is really important. Furthermore, the increase in basal ATPase activity and impact for LTC4 binding for the double R928A and K932A mutant seems minimal. Therefore, the authors are encouraged to address the R domain's role in WT context.

4. Given the hypothesis for the role of R domain in regulating substrate binding and ATP hydrolysis, it is puzzling to see that the structure of the OF conformation was obtained under the autoinhibited conditions with a substrate concentration of 40 μM that is supposedly not able to relieve the inhibition. It appears to this reviewer that using the E/Q mutant with ATP and 40 μM LTC4, two structures were obtained: one is the auto-inhibited IF and another is the OF with both NBDs occupied with ATP. This result appears in direct contradiction to the hypothesis that only high concentration of LTC4 is able to relieve autoinhibition of R domain and the interaction of R domain (927-934) with ATP bound at NBD2 provides additional regulation of ATP hydrolysis.

Questions:

1. Did authors try to figure out the phosphorylation state of the purified MRP2 protein, since phosphorylation has been reported to regulate MRP2 function? Both Mazza et al. and Mao et al. report that their recombinant MRP2 proteins are partially phosphorylated after purification at several predicted sites using Mass Spec.
2. Line 100: Are the authors confident on assigning the extra densities as lipid and cholesterol at the interface between TMD0 and TMD1? If it's just a hypothesis (not sure due to low local resolution), then it's best to state this observation as a "possibility" or predicted to be.
3. As the authors mentioned that the wildtype and mutant MRP2 in the autoinhibited state show difference in the amount of ordered R domains, how much of a difference is there in terms of the number of residues modeled? How is the wild type R domain (this work) compared to those of the published work?
4. In Figure 2D, the unresolved hydrophobic tail of LTC4 is shown in gray but is surrounded by W1254, T1251, S1101, I547, and F550. The authors should explain how they determined that this tail region interacts with these five amino acids, as these interactions weren't observed in their structure. Labeling these amino acids with different colors could help clarify this.
5. Does the scrambled R domain allow transport of substrate at lower concentration?

Minor corrections:

1. L178 and legend of Figure 2H: The K_m values for LTC4: WT: 6.3 μM ; R928A/K932A: 11.2 μM ; and Scramble: 0.6 μM . But Figure 2H labels LTC4 concentration in mM.
2. The Lasso loop in Figures 1A and 1B could be colored differently to make it easier to recognize.
3. Not all residues displayed in Figure 1D are mentioned in the text. More detailed should be provided for these residues in the article.
4. The TM6, 8, 16, and 17 in Figure 2C can be labeled, and more detail should be provided in lines 147 and 148.
5. Since the authors used R928A/K932A and Scramble mutants for ATP hydrolysis activity study (Figures 2G and 2H), it would be helpful if they describe the interaction between the R domain and ATP (lines 123 to 129) with reference to Figure 1E.
6. The MRP(E1462) in Supplementary Table 1 should be corrected to MRP(E1462Q).
7. Figures 1E, 3B, Supplementary Figures 3D, 3F, 4D and 6: the contour level of the density is not stated.
8. L296: and the temperature was dropped to from 37 $^{\circ}\text{C}$ to 30 $^{\circ}\text{C}$
9. Line 107: residue 861-893 will need to show either electron density or local resolution map to prove the resolution is high enough to see it's a beta sheet.

10. The figure caption of S3F needs clarification that it is the ATP bound at the NBD1.

11. Figure S4D, please label the surrounding residues. Otherwise, it's difficult to tell where the binding pocket is located, although it's clearly stated in the main text.

12. Figure 2C: please show inhibitor density in the main figure like in S4D.

Reviewer #3

(Remarks to the Author)

Reviewer Comments

In this paper, Koide et al. demonstrated three structures of human MRP2, autoinhibited state, substrate-bound pre-translocation state, and ATP-bound post-translocation state. In addition, this manuscript provided insights into the MRP2 transport cycle and an autoinhibited conformation. The findings suggested that the R domain functions as an auto-inhibitor to down-regulate ATP hydrolysis, and substrate release occurs before ATP hydrolysis. Overall, this manuscript is well-structured with high writing quality. I would recommend acceptance of the paper after the authors successfully address the following comments.

1. For Page 3, lines 40-43, please list some details of these MRP2 knockout mice.
2. In the introduction section, it would be better if there were more descriptions or explanations of the ABC transporter superfamily, for example, its classification, as well as the basic structures of other ABC transporters.
3. For Figure 1, the authors should add some figure legends indicating different domains are colored differently, as well as some interpretations if the colors have different meanings.
4. For Page 4, lines 63-70, it would be better if the authors added a figure indicating the structure of ABCB1, ABCG2, and MRP1.
5. It would be interesting to see some discussion about some other ABC transporters, such as MRP7, MRP8, and MRP9, in the introduction section.
6. The authors should update the citations if possible. It would be better if the references were up-to-date as to reflect the current progress of the field unless containing key findings.
7. In the discussion section, the authors mention that two similar studies were published prior to this study. How do the results obtained from these three independent studies contribute to the understanding of the MRP2 transport mechanism? Please cite and discuss some data from the previous two studies.
8. The authors should consider providing more background information and evidence on MRP2 and its importance in multidrug resistance.
9. It would be interesting to see the authors test more substrate drugs that can be transported by MRP2. If the authors have tested other substrates, please provide these data.

Reviewer #4

(Remarks to the Author)

Version 1:

Reviewer comments:

Reviewer #1

(Remarks to the Author)

This revision is a disappointing update to the original text. Based on the responses to reviewer comments, it appears that the authors have made improvements to the writing of the manuscript but very little to expand or strengthen the data or understanding of the binding pocket, despite being given feedback from more than one reviewer on the disconnect between what was promised in the title and abstract and what was delivered in the text. The two previous papers, also published in Nature Communications, presented an in-depth analysis regarding how ABCC2 activity was modulated by phosphorylation or mutagenesis. Although supplemental text has been added in the discussion section, the analysis of ligand interactions is still largely superficial, and no true structure-activity relationships have been probed or explored in the way presented in the previous two papers. Additionally, the discussion of the probenecid-bound structure is very misleading if not incorrect and does not fit with the extensive known transport biochemistry. From the text, probenecid is largely presented as a substrate of MRP2 ("Comparative structural analyses of MRP2 bound to various substrates", line 28; "Despite being much smaller than typical MRP2 substrates, two copies of Probenecid occupy the binding site to facilitate its efflux by MRP2," line 277; "Figure 5. Structural basis of multi-substrate recognition," line 578) despite that two references prominently cited in the manuscript, Mazza et al (reference 30) and Bakos et al (reference 31), clearly present that probenecid can both inhibit or modulate MRP2 activity. In addition, Mazza et al describe their hypothesis as to how the binding pocket may be occupied differently in the

case of probenecid inhibition vs. modulation, which is omitted from the authors' analysis of the three ligand-bound MRP2 structures. Mazza et al hypothesizes that when acting as an inhibitor, two copies of probenecid occupy the binding site, whereas when acting as a modulator, one molecule of probenecid may be displaced from the less conserved drug binding site 1. Based on this hypothesis, the structure compared in this manuscript corresponds to the structure of probenecid acting as an inhibitor. The authors ignored a key opportunity to compare and contrast the differences between interacting residues with probenecid, as an inhibitor or modulator, and LTC4 and BDT, as substrates, ideally introducing mutations or new ligands to test their hypotheses via a traditional structure-activity relationship studies. The authors could have also investigated evolutionary relationships, which they only briefly discussed comparing the Y to F point mutation between MRP1 and MRP2. Another avenue to explore would be looking for evolutionarily conserved residues between species for broad insight on how MRP2 binds its ligands. Although the authors have added a phrase to the abstract to point out that they are comparing their structure to previously published structures, it is still not obvious, especially looking at the figures. Taking the misleading analysis of Mazza et al's probenecid-bound structure aside, this reviewer does not feel that the currently available new data, as it is presented in this manuscript, is a significant enough advance beyond the previous papers.

Chemical structures described as "tan" in figure 5B, D, and F do not appear tan. Please update color to brown or burgundy.

Reviewer #2

(Remarks to the Author)

Comments on the revised manuscript entitled "Structural basis for the transport and regulation mechanism of the Multidrug resistance associated protein 2" by Koide et al.

The authors did a good job providing clarifications on various questions raised by reviewers. That said, the role for the second site interaction of the R domain remains elusive. The question is what role this particular interaction, enhanced/stabilized by the E/Q mutation at the NBS2 as it is not observed in wt structures, may play for the function of ABCC2.

This reviewer also puzzled over the contradictory observations that both the prehydrolysis state (ATP with E/Q mutation in NBS2) and the posthydrolysis state (ADP in wt NBS2, Mao et al.) have a near identical conformation. This issue seems to be avoided in the discussion especially in terms of whether the use of E/Q mutation is appropriate in providing an accurate mechanistic picture in this case.

Reviewer #3

(Remarks to the Author)

The authors had addressed the comments accordingly. It is acceptable for publication

Reviewer #4

(Remarks to the Author)

Version 2:

Reviewer comments:

Reviewer #3

(Remarks to the Author)

I think that the explanations from the authors regarding probenecid and MRP2 is reasonable. I do not have additional comments.

Point-by-point Responses

We thank the reviewers for their time and constructive comments, which helped us to improve the manuscript substantially. Our point-by-point responses are shown in blue ink.

Reviewer #1 (Remarks to the Author):

- This manuscript is overall well-written and easy to follow. However, there are major edits required for the work as it stands. The manuscript reads as if it was started before the first human ABCC2 structures were reported, and then later the authors pivoted to discussing their structures along with the previously reported structures. Although there are definitely novel data presented in this manuscript, the authors are not convincing in championing their work as a contribution to the field. For example, the authors report a previously unresolved portion of the R-domain which is absent from previously reported structure which reaches onto the NBD2 surface to form a composite ATP-binding site. This is a really interesting finding, especially given that NBD2 is the consensus site, but it's barely highlighted or discussed. Similarly, the analysis of¹ the binding pocket based on their LTC4-bound structure and previously reported BDT- and probenecid-bound structure is cursory. This manuscript would be greatly strengthened by the elaboration of the discussion and analysis of these two topics.

We have now revised the abstract and the text to expand upon these two topics. For example, in the abstract, we now included: *"In the autoinhibited state, the cytosolic regulatory (R) domain plugs into the transmembrane substrate-binding site and extends into the cytosol to form a composite ATP-binding site at the surface of nucleotide-binding domain 2"*. We have also revised the text on pages 6-7 and 11-12 to discuss the mechanistic implications of the novel ATP-binding site and to compare different substrate-bound structures.

- Although it is not explicitly required, for consistency naming using the HUGO gene nomenclature is recommended for this manuscript. Currently, the authors use a mix of HUGO and common names for the transporters (MRP2, MRP1, P-gp, ABCG2). The authors do state P-gp is also known as ABCB1, however MRP2 and MRP1 are never defined as ABCC2 and ABCC1, respectively. It is recommended to identify all transporters as their HUGO Gene Nomenclature Committee (HGNC) names, or at the very least to include both.

We have revised the manuscript to include HUGO gene nomenclature for all ABC transporters discussed in the manuscript.

- There is a bit of a disconnect between the abstract and the manuscript submitted. Line 28 (abstract): "Comparative structural analyses of MRP2 in complex with different substrates reveal how the transporter recognizes a diverse array of compounds, highlighting the transporter's role in multidrug resistance." The wording of this phrase in the abstract suggests that this manuscript will present structures of ABCC2 with different substrates. However, only one substrate-bound structure is presented, and the comparison is performed using previously derived structures from other groups. This needs modification to more accurately reflect what was evaluated in the current manuscript. In a similar vein, none of the substrates bound to ABCC2 in these structures are chemotherapeutics, so it seems more appropriate to say that this manuscript discusses the polyspecificity or promiscuity of ABCC1, which supports its role in multidrug resistance.

Thank you. The abstract is now revised as: *"Comparative structural analyses of MRP2 bound to various substrates, as determined in this study and others, reveal how the transporter recognizes a diverse array of compounds, supporting its role in multidrug resistance"*.

- Lines 171-175: "Disruption of the interactions with the substrate-binding site led to a more pronounced change. Replacement of the entire R domain (residues 861-960) with a scrambled amino acid sequence ("scramble") increased the basal ATPase rate by more than 10-fold (Fig. 2G). These results suggest that by disrupting the interactions between the R domain and the transporter core, the R domain is more readily released from its auto-inhibitory role, leading to an elevation in the ATPase activity even in the absence of substrate." Given that the entire R-domain sequence has been scrambled, you have in essence mutated the R-

domain to something that is not an R-domain. I agree with your conclusion that the ATPase activity is increased. However, because the R-domain is absent, it seems inconceivable that a completely scrambled version of this domain is still able to properly insert and release from within the TMDs. Please verify this structurally. Also, because the ATPase changes so strongly, please determine if this is due to a change in the NBD dimerization and/or change in affinity for ATP.

We agree that the scrambled version is unlikely to insert into the intracellular opening, therefore losing its inhibitory function, leading to the 10-fold increase of ATPase activity. We have revised the text to reflect this view: *“The scramble variant, which likely disrupts most or all interactions with the transporter core, led to a more pronounced effect”*.

To assess whether the affinity for ATP has changed, we determined the K_m for ATP for both the WT and scrambled variant. The data show that wild-type MRP2 hydrolyzes ATP with a K_m of 0.35 ± 0.03 mM (mean \pm standard error), while the scrambled version has a K_m of 0.23 ± 0.02 mM. These data suggest that the change in ATPase activity for the scrambled variant is largely due to the increased rate of NBD dimerization rather than a difference in affinity for ATP. We have now included these data in Figure 1G and revised the text accordingly.

- Lines 199-200: “In particular, K953, Q955, and G960 of the R domain form hydrogen bonds with R1079 and R1083 of TM helix 14 and with residue R1150 of TM helix 15.” K953 is not shown in Figure 3 and the 955 residue is labeled as E955, not Q955. I believe this is a typo in the text.

We corrected the typo and revised the figure and text to be consistent. It now reads: *“These residues dock along the intracellular region of the TM helices, stabilized by a network of van der Waals interactions and hydrogen bonds formed between E955 and R1083, as well as between the mainchain atoms of K953 and G960 and the side chains of R1079 and R1150, respectively (Fig. 3A,B)”*.

- Lines 223-226: “All three studies report a similarly inward-facing, apo conformation with the R domain inserted into the transmembrane substrate-binding site. In addition, we observed the C-proximate region of the R domain that, in the presence of ATP, reaches onto the NBD2 surface to form a composite ATP-binding site (Fig. 1).” This section of the R-domain is unresolved in previously reported ABCC2 structures, correct? It seems like this discrepancy should be discussed more prominently.

We thank the authors for this comment and have elaborated on this description in the text as the following: *“In addition, we observed a previously unresolved region of the R domain, residues 927-934, reaches onto the NBD2 surface to form a composite ATP-binding site (Fig. 1). Given that MRP2 hydrolyzes ATP with a K_m of 0.35 mM and that intracellular ATP concentrations range between 1 and 10 mM, the ATP-bound conformation observed in this study is likely to represent the physiological state of MRP2. In this conformation, the R domain not only occupies the substrate-binding site in the TMDs, it also sequesters the ATP-binding site of NBD2”*.

- Line 237: “Chen and colleagues” is used for reference 24, which corresponds to the corresponding author, not the first author, Mao. Please modify.

We have made the correction.

- Line 264 through end of discussion: “The negative moieties of each compound form hydrogen bonds with the same set of residues in the P-pocket (Fig. 5).” The analysis provided in this paragraph is quite broad and vague. These statements would be perfectly appropriate if the binding pocket was discussed more in depth previously, either when discussing the LTC4-bound structure or in a dedicated section preceding the discussion. However, the discussion of the promiscuity of the binding pocket is largely found wanting.

As suggested, we have expanded the description of substrate interactions and the discussion of MRP2 promiscuity on page 12.

- Methods section: There are multiple instances of a space included between a given number and “°C.” Please

revise so that, for example, “30 °C” becomes “30°C.”

We have removed the space in all instances.

- Line 319-320: “New England Biotechnologies, NEB.” I believe NEB stands for New England Biolabs.

We have made the correction.

- Line 340: “ $\lambda_{ex}/\lambda_{em}$ ” requires subscripts

We reformatted as suggested.

- Figure 2D, 5B,D, and F, and S5: Each of these figures includes at least one panel with schematic drawings where amino acid residues in colored ovals are shown interacting with chemical structures of ligands. In figure S5, the coloring is well explained (P-pocket in blue, H-pocket in gray). However, this panel is the only one with adequate details to interpret the figure. Figure 2D shows ovals in blue and green, which I have interpreted as referring to which TMD the residues come from, as is colored in Figure 2A-C. However, the caption must explain what the color scheme means for figure 2D. I am completely lost on Figure 5 B, D, and F, which contain purple and gray ovals. The coloring does not correlate with the numbering as observed in Figure 2D, where TMD1 residues (#329-591) are in green and TMD2 (#1101-1257) are in blue. The coloring also does not correlate with the P-pocket and H-pocket residues from figure S5. Please either update the caption to explain why the residues are colored differently or update the coloring and caption to mirror figure 2D or S5.

We apologize for the lack of clarity and have updated figure legends with descriptions of the color scheme.

Reviewer #2 (Remarks to the Author):

Comments on the manuscript entitled "Structural basis for the transport and regulation mechanism of the Multidrug resistance associated protein 2" by Koide et al.

In this work, the authors reported four cryo-EM structures of human MRP2 in three conformational states: an autoinhibited state for the wild-type in the absence of ligand and ATP (3.6 Å) and for the E/Q mutant in the absence of ligand but in the presence of ATP (3.4 Å), a pre-translocation state for the wild-type with substrate-bound (LTC4) in the absence of ATP (2.9 Å), and an ATP-bound post-translocation state for the E/Q mutant (3.4 Å). The authors reported the observation of large-scale conformational changes that enable ATP-dependent substrate translocation, showing that MRP2 functions through the classic alternating access model, driven by ATP binding and hydrolysis.

The EM density for the regulatory R domain was found to be traceable for a large portion of the domain, leading to a revelation of its two interaction sites with other parts of MRP2. In addition to the structural work, two mutants were made in the R domain and analyzed for their importance to the observed interactions. The R domain was proposed to serve as a selectivity gauge, wherein only sufficiently high concentrations of substrates or substrate with sufficiently high affinity can effectively compete with and disengage the R domain to initiate transport.

The authors performed cryoEM experiments and structure determination carefully. The cryoEM data processing flow charts illustrated in supplementary figures did not show any sign of discrepancy. For example, the FSC curves are acceptable for C1 reconstruction, and the 2D class averages and final reconstructions with local resolution maps are all within expectation.

Although this reviewer recognizes past important contributions of the group to the structural studies of ABC transporters in general and ABCC family proteins in particular, there are several concerns with the current submission.

Major concerns:

1. The results reported here are consistent with the two recent publications in January and February of this

year in Nature Communications by Mazza et al. (Reference 24 for the rat MRP2 structures in nucleotide-free autoinhibited state and probenecid bound state) and by Mao et al. (References 25, human wild type MRP2 in apo state, BDT bound state, and ATP/ADP state). In addition to their observation of the insertion of the R domain into the substrate-binding site in the TMD, these two papers also demonstrated regulatory effect of R domain on MRP2's ATPase and transport activities using mutagenesis, phosphomimetics, and proteins that are fully dephosphorylated, partially and fully phosphorylated. Thus, the new information presented in this submission appears to limit to an additional interaction between the C-proximate region of the R domain and the ATP bound at the NBD2 in the IF conformation. To enhance its novelty, the authors should provide additional data in the manuscript.

Compared to the published work, we present here two additional conformations that were not described before: the autoinhibited, ATP-bound structure in which a novel ATP-binding site is formed between the R domain and NBD2, and the ATP-bound, NBD-dimerized structure representing the pre-hydrolysis state. Additionally, we also had the opportunity to compare the structures of MRP2 bound with three different substrates to understand the origin of its "multidrug" recognition. For these reasons, we believe this work ties well with the two recent publications to form a cohesive body of knowledge regarding the fundamental mechanism of MRP2.

2. The title of the manuscript is vague enough to be an umbrella under which this work might be presented if it wasn't for statements like "different substrates reveal how the transporter recognized a diverse array of compounds, highlighting the transporter's role in multidrug resistance" (Abstract lines 29-30). Although the discussion on the structural basis of multi-substrate recognition is interesting, it is based largely on results from the two recent publications.

We have revised the abstract to highlight the novel ATP-binding site identified in this study.

3. Like the two recent publications, the focus of this manuscript is about the R domain, especially the part of R domain from residues 927-934 that is found in this work interacting with ATP bound at the NBD2. The authors imply that this interaction, in the absence of substrate, prevents MRP2 from hydrolyzing ATP, providing another level of regulation in addition to the universally observed interaction at the substrate-binding site. Therefore, the R domain plays a role as an off switch in resting state in order to conserve ATP in cells. However, this interaction of R domain (927-934) with ATP bound in NBD2 can only be observed in the E1462Q mutant instead of WT, calling into question whether this interaction is really important. Furthermore, the increase in basal ATPase activity and impact for LCT4 binding for the double R928A and K932A mutant seems minimal. Therefore, the authors are encouraged to address the R domain's role in WT context.

When ATP was included with the WT protein, the sample became heterogeneous due to the basal ATPase activity of MRP2. The E1462Q mutation prevents ATP hydrolysis by eliminating the catalytic base without altering the structure of the ATP-binding site. The equivalent mutation has been used in many ABC transporters to analyze ATP binding at the NBD dimer interface. In the inward-facing conformation, the E1462Q residue of MRP2 does not make any contact with the R domain or ATP, so it is very unlikely to cause any artifacts.

The modest effect of the R928A/K932A mutation is expected, as the interface between the R domain and the transporter core is extensive. Point mutations are likely to have a limited effect in disrupting the R domain insertion. The fact that we observed a statistically significant change in this mutant (Figure 1F) strongly supports the physiological relevance of this interface.

4. Given the hypothesis for the role of R domain in regulating substrate binding and ATP hydrolysis, it is puzzling to see that the structure of the OF conformation was obtained under the autoinhibited conditions with a substrate concentration of 40 μ M that is supposedly not able to relieve the inhibition. It appears to this reviewer that using the E/Q mutant with ATP and 40 μ M LCT4, two structures were obtained: one is the auto-inhibited IF and another is the OF with both NBDs occupied with ATP. This result appears in direct contradiction to the hypothesis that only high concentration of LCT4 is able to relieve autoinhibition of R domain and the interaction of R domain (927-934) with ATP bound at NBD2 provides additional regulation of ATP hydrolysis.

We have a different interpretation. In the absence of substrate, MRP2 does hydrolyze ATP, albeit at a low rate. This basal activity is caused by the infrequent disengagement of the R domain, which permits NBD dimerization. Therefore, an OF (open-facing), NBD-dimerized state exists even without substrate, but its low abundance typically precludes structural observation. Substrate accelerates the rate of R domain disengagement and NBD dimerization. The E/Q mutant prevents ATP hydrolysis and stabilizes the OF conformation. Therefore, even with 40 μ M LTC4, the probability of the OF conformation was sufficiently increased to permit structural determination.

Questions:

1. Did authors try to figure out the phosphorylation state of the purified MRP2 protein, since phosphorylation has been reported to regulate MRP2 function? Both Mazza et al. and Mao et al. report that their recombinant MRP2 proteins are partially phosphorylated after purification at several predicted sites using Mass Spec.

Yes, we did test this. The purified protein sample was phosphorylated, as indicated by the phosphoprotein gel stain (Figure R1, right panel, lane 4). Lambda phosphatase reduced phosphorylation (Figure R1, lane 1), and incubating the dephosphorylated sample with kinases Casein Kinase II (CKII) or Protein Kinase A (PKA) both increased the level of phosphorylation (Figure R1, lanes 2 and 3). We are currently investigating whether and how phosphorylation regulates MRP2's activity.

Figure R1: Phosphorylation test

Left: Purified MRP2, stained with Coomassie brilliant blue (left).

Right: Same sample as the right panel, stained with Pro-Q Diamond phosphoprotein gel stain (right).

MRP2 dephosphorylated using lambda phosphatase (lane 1), dephosphorylated MRP2 treated with PKA (lane 2), dephosphorylated MRP2 treated with CKII (lane 3), untreated MRP2 (lane 4).

2. Line 100: Are the authors confident on assigning the extra densities as lipid and cholesterol at the interface between TMD0 and TMD1? If it's just a hypothesis (not sure due to low local resolution), then it's best to state this observation as a "possibility" or predicted to be.

The densities were very strong and consistent with the shape of cholesterol and lipids. We reworded to reflect uncertainty in interpreting cryo-EM maps. It now reads: "In the transmembrane region, TMD0 and the transporter core have few direct contacts, with densities likely representing cholesterol and lipid molecules filling the gap between them".

3. As the authors mentioned that the wildtype and mutant MRP2 in the autoinhibited state show difference in the amount of ordered R domains, how much of a difference is there in terms of the number of residues modeled? How is the wild type R domain (this work) compared to those of the published work?

The WT MRP2 contains residues 886-915 of the R domain, the E/Q mutant contains residues 860-935. The published structure, PDB ID 8jy5, contains residues 860-914, followed by a backbone model of residues 915-932.

4. In Figure 2D, the unresolved hydrophobic tail of LTC4 is shown in gray but is surrounded by W1254, T1251, S1101, I547, and F550. The authors should explain how they determined that this tail region interacts with these five amino acids, as these interactions weren't observed in their structure. Labeling these amino acids with different colors could help clarify this.

We agree that Figure 2D is misleading as the exact interactions with the lipid tail is uncertain. We have now deleted it in the revision.

5. Does the scrambled R domain allow transport of substrate at lower concentration?

Although we did not directly measure substrate transport, ATPase assays showed that the scramble variant interacts with LTC₄ with an affinity 10-fold higher than the WT, suggesting that this variant may permit substrate transport at lower concentrations than the WT protein (Figure 2F).

Minor corrections:

1. L178 and legend of Figure 2H: The K_m values for LTC₄: WT: 6.3 μM; R928A/K932A: 11.2 μM; and Scramble: 0.6 μM. But Figure 2H labels LTC₄ concentration in mM.

We thank the reviewer for identifying this error in Figure 2H labels. We have now updated the label to reflect micromolar values.

2. The Lasso loop in Figures 1A and 1B could be colored differently to make it easier to recognize.

As suggested, we have updated Figure 1A and 1B as well as 2A and 3A to use a darker yellow color for the lasso motif.

3. Not all residues displayed in Figure 1D are mentioned in the text. More detailed should be provided for these residues in the article.

We have revised the text to discuss all the residues highlighted in Figure 1D. It now reads: “Near the apex of the TM cavity, the loop-turn-helix motif of the R domain engages in hydrogen bonding and extensive van der Waals interactions with TMD1 and TMD2 (Fig. 1D). Many residues at this interface, including R1205, N1204, R1257, W1254, and N332, are involved in substrate binding (discussed later, in Fig. 2). Additionally, the helical region of the R domain forms contacts with both halves of the transporter, through a salt bridge formed between R905 and D433 in the first half and a hydrogen bond between R910 and Q488 of the second half (Fig. 1D)”.

4. The TM6, 8, 16, and 17 in Figure 2C can be labeled, and more detail should be provided in lines 147 and 148.

We have now revised Figure 2 and the text to describe the binding site in greater detail.

5. Since the authors used R928A/K932A and Scramble mutants for ATP hydrolysis activity study (Figures 2G and 2H), it would be helpful if they describe the interaction between the R domain and ATP (lines 123 to 129) with reference to Figure 1E.

We have now revised Figure 1 to include the ATP hydrolysis data and revised the text to tie the structure to function. Thank you for this suggestion.

6. The MRP(E1462) in Supplementary Table 1 should be corrected to MRP(E1462Q).

We have made the correction.

7. Figures 1E, 3B, Supplementary Figures 3D, 3F, 4D and 6: the contour level of the density is not stated.

We have updated the figure legends of all figures containing cryo-EM density to include the contour level of the density.

8. L296: and the temperature was dropped to from 37 °C to 30 °C

We have made the correction.

9. Line 107: residue 861-893 will need to show either electron density or local resolution map to prove the resolution is high enough to see it's a beta sheet.

We have updated Fig. S3E to include electron density for this region.

10. The figure caption of S3F needs clarification that it is the ATP bound at the NBD1.

We have revised to include the clarification.

11. Figure S4D, please label the surrounding residues. Otherwise, it's difficult to tell where the binding pocket is located, although it's clearly stated in the main text.

We have updated figure S4D to include the surrounding residues and the model of LTC4 (excluding its unresolved hydrophobic tail).

12. Figure 2C: please show inhibitor density in the main figure like in S4D.

Density for LTC4 is shown in figure S4D. We believe adding the same density in Figure 2C will make it difficult to appreciate the protein-ligand interaction.

Reviewer #3 (Remarks to the Author):

Reviewer Comments

In this paper, Koide et al. demonstrated three structures of human MRP2, autoinhibited state, substrate-bound pre-translocation state, and ATP-bound post-translocation state. In addition, this manuscript provided insights into the MRP2 transport cycle and an autoinhibited conformation. The findings suggested that the R domain functions as an auto-inhibitor to down-regulate ATP hydrolysis, and substrate release occurs before ATP hydrolysis. Overall, this manuscript is well-structured with high writing quality. I would recommend acceptance of the paper after the authors successfully address the following comments.

1. For Page 3, lines 40-43, please list some details of these MRP2 knockout mice.

The description of the MRP2 knockout mice now reads: "*MRP2 knockout mice exhibit hyperbilirubinemia, reduced bile flow, reduced biliary glutathione excretion, and an increase in liver size*¹⁰. These data indicate that MRP2 plays a role in maintaining liver homeostasis by excreting potentially toxic molecules" and "*MRP2 inhibition is associated with improved efficacy of cisplatin sensitivity in hepatocellular carcinoma, and MRP2 knockout mice exhibit decreased excretion of the chemotherapeutics methotrexate and doxorubicin*^{10,13}".

2. In the introduction section, it would be better if there were more descriptions or explanations of the ABC transporter superfamily, for example, its classification, as well as the basic structures of other ABC transporters.

We have revised the introduction to include MRP2's classification as a member of the ABCC subfamily, and the canonical ABC transporter structure, which consists of two transmembrane domains (TMD1 and TMD2) and two nucleotide-binding domains (NBD1 and NBD2).

3. For Figure 1, the authors should add some figure legends indicating different domains are colored differently, as well as some interpretations if the colors have different meanings.

We revised the Figure 1 legend as suggested.

4. For Page 4, lines 63-70, it would be better if the authors added a figure indicating the structure of ABCB1, ABCG2, and MRP1.

This section is intended to briefly summarize the known mechanisms of these transporters, which may not be fully conveyed in a single figure. Instead, we have listed all the references for interested readers to explore further.

5. It would be interesting to see some discussion about some other ABC transporters, such as MRP7, MRP8, and MRP9, in the introduction section.

Due to space concerns, we prefer to focus on MRP2 and provide only a brief summary of the relevant information regarding other ABC transporters.

6. The authors should update the citations if possible. It would be better if the references were up-to-date as to reflect the current progress of the field unless containing key findings.

We have updated citations as suggested.

7. In the discussion section, the authors mention that two similar studies were published prior to this study. How do the results obtained from these three independent studies contribute to the understanding of the MRP2 transport mechanism? Please cite and discuss some data from the previous two studies.

As suggested, on pages 10-12, we discuss how these three studies complement each other to provide a better understanding of the functional role of the R domain and how MRP2 recognizes different substrates.

8. The authors should consider providing more background information and evidence on MRP2 and its importance in multidrug resistance.

We have revised the introduction to highlight the functional importance of MRP2, including studies by Qu et al., which show that inhibition of MRP2-mediated chemotherapeutic export may enhance chemosensitivity in hepatocellular carcinoma, and by Myint et al., which show that MRP2 is differentially expressed in gastrointestinal cancers of patients who did not respond to FOLFOX chemotherapy.

9. It would be interesting to see the authors test more substrate drugs that can be transported by MRP2. If the authors have tested other substrates, please provide these data.

We agree that understanding the full substrate spectrum of MRP2 is very important, but we believe it is beyond the scope of this manuscript.

Point-to-point response

Reviewer #1 (Remarks to the Author):

This revision is a disappointing update to the original text. Based on the responses to reviewer comments, it appears that the authors have made improvements to the writing of the manuscript but very little to expand or strengthen the data or understanding of the binding pocket, despite being given feedback from more than one reviewer on the disconnect between what was promised in the title and abstract and what was delivered in the text. The two previous papers, also published in Nature Communications, presented an in-depth analysis regarding how ABCC2 activity was modulated by phosphorylation or mutagenesis. Although supplemental text has been added in the discussion section, the analysis of ligand interactions is still largely superficial, and no true structure-activity relationships have been probed or explored in the way presented in the previous two papers.

Additionally, the discussion of the probenecid-bound structure is very misleading if not incorrect and does not fit with the extensive known transport biochemistry. From the text, probenecid is largely presented as a substrate of MRP2 (“Comparative structural analyses of MRP2 bound to various substrates”, line 28; “Despite being much smaller than typical MRP2 substrates, two copies of Probenecid occupy the binding site to facilitate its efflux by MRP2,” line 277; “Figure 5. Structural basis of multi-substrate recognition,” line 578) despite that two references prominently cited in the manuscript, Mazza et al (reference 30) and Bakos et al (reference 31), clearly present that probenecid can both inhibit or modulate MRP2 activity. In addition, Mazza et al describe their hypothesis as to how the binding pocket may be occupied differently in the case of probenecid inhibition vs. modulation, which is omitted from the authors’ analysis of the three ligand-bound MRP2 structures. Mazza et al hypothesizes that when acting as an inhibitor, two copies of probenecid occupy the binding site, whereas when acting as a modulator, one molecule of probenecid may be displaced from the less conserved drug binding site 1. Based on this hypothesis, the structure compared in this manuscript corresponds to the structure of probenecid acting as an inhibitor. The authors ignored a key opportunity to compare and contrast the differences between interacting residues with probenecid, as an inhibitor or modulator, and LTC4 and BDT, as substrates, ideally introducing mutations or new ligands to test their hypotheses via a traditional structure-activity relationship studies. The authors could have also investigated evolutionary relationships, which they only briefly discussed comparing the Y to F point mutation between MRP1 and MRP2. Another avenue to explore would be looking for evolutionarily conserved residues between species for broad insight on how MRP2 binds its ligands. Although the authors have added a phrase to the abstract to point out that they are comparing their structure to previously published structures, it is still not obvious, especially looking at the figures. Taking the misleading analysis of Mazza et al’s probenecid-bound structure aside, this reviewer does not feel that the currently available new data, as it is presented in this manuscript, is a significant enough advance beyond the previous papers.

We propose a different interpretation regarding the function of probenecid and the probenecid-bound structure reported by Mazza et al. ABC transporters typically use ATP hydrolysis to drive substrate translocation, with substrate presence often enhancing ATPase activity. Both Mazza et al. (reference 30) and Bakos et al. (reference 31) showed that instead of inhibiting, probenecid increases the ATPase activity of MRP2, with the cryo-EM structure (reference 30) revealing probenecid bound at the same site as other substrates. This structure was determined in the presence of 1 mM probenecid (reference 30), a concentration at which Mazza et al. also

observed an increase in MRP2's ATPase activity. Therefore, we interpret the cryo-EM structure as representing a substrate-bound conformation, rather than an inhibitory state.

If probenecid is simply a substrate of MRP2, how does it inhibit the transport of NEM-GS (reference 31) or CDF (reference 30)? We believe this functional effect occurs not through inhibition of MRP2 but rather through competitive binding. This mechanism is reminiscent of the perchlorate inhibition of thyroidal iodine accumulation. As described by Dr. Nancy Carrasco, contrary to previous belief, perchlorate is not an inhibitor of the transporter but is actually translocated by the Na⁺/I⁻ symporter as a substrate (Dohán O, et al., 2007, PMID: 18077370). We believe that probenecid acts in a similar manner on MRP2 to inhibit the transport of other substrates through competition.

Chemical structures described as “tan” in figure 5B, D, and F do not appear tan. Please update color to brown or burgundy.

We have revised the color scheme in Figure 5 as suggested. We also updated Figure 5 with the PDB codes and the legend to clearly indicate that the BDT- and probenecid-bound structures are based on previous publications.

Reviewer #2 (Remarks to the Author):

Comments on the revised manuscript entitled "Structural basis for the transport and regulation mechanism of the Multidrug resistance associated protein 2" by Koide et al.

The authors did a good job providing clarifications on various questions raised by reviewers. That said, the role for the second site interaction of the R domain remains elusive. The question is what role this particular interaction, enhanced/stabilized by the E/Q mutation at the NBS2 as it is not observed in wt structures, may play for the function of ABCC2.

Because perturbing this interaction with mutations in the background of wild-type MRP2 caused a statistically significant increase in basal ATPase activity (Figure 1F), we believe the functional role of this interaction is to stabilize the R domain in the inhibitory position at physiological ATP concentrations.

This reviewer also puzzled over the contradictory observations that both the prehydrolysis state (ATP with E/Q mutation in NBS2) and the posthydrolysis state (ADP in wt NBS2, Mao et al.) have a near identical conformation. This issue seems to be avoided in the discussion especially in terms of whether the use of E/Q mutation is appropriate in providing an accurate mechanistic picture in this case.

The structural similarity between these two conformational states has significant mechanistic implications. We previously detailed this phenomenon in studies of MRP1, a closely related transporter. Like MRP2, MRP1 exhibits identical structures in the ATP-bound, pre-hydrolysis state and the ATP/ADP-bound, post-hydrolysis state, indicating that ATP hydrolysis *per se* does

not reset MRP1 to the resting state. Supporting this, smFRET studies have shown that the rate-limiting step in the MRP1 transport cycle is the dissociation of the NBD dimer. Given the close relationship between MRP1 and MRP2, it is unsurprising that they share this transport mechanism. We have now revised the discussion to reference relevant MRP1 studies: “The nearly identical structures of these two conformations indicates that, similar to MRP1^{25,26}, substrate is released before ATP hydrolysis, and dissociation of the NBD dimer likely constitutes the rate-limiting step in the transport cycle”.

References:

25. Johnson, Z. L. & Chen, J. ATP Binding Enables Substrate Release from Multidrug Resistance Protein 1. *Cell* **172**, 81-89.e10 (2018).
26. Wang, L. *et al.* Characterization of the kinetic cycle of an ABC transporter by single-molecule and cryo-EM analyses. *eLife* **9**, e56451.

Reviewer #3 (Remarks to the Author):

The authors had addressed the comments accordingly. It is acceptable for publication

Reviewer #4 (Remarks to the Author):
